# Characterization and Antibiofilm Activity of Mannitol–Chitosan-Blended Paste for Local Antibiotic Delivery System

**DOI:** 10.3390/md17090517

**Published:** 2019-09-02

**Authors:** Leslie R. Pace, Zoe L. Harrison, Madison N. Brown, Warren O. Haggard, J. Amber Jennings

**Affiliations:** Department of Biomedical Engineering, Herff College of Engineering, University of Memphis, Memphis, TN 38152, USA

**Keywords:** biofilm, chitosan, biomaterial, local drug delivery, persister, *Staphylococcus*, infection, mannitol, implant

## Abstract

Mannitol, a polyalcohol bacterial metabolite, has been shown to activate dormant persister cells within bacterial biofilm. This study sought to evaluate an injectable blend of mannitol, chitosan, and polyethylene glycol for delivery of antibiotics and mannitol for eradication of *Staphylococcal* biofilm. Mannitol blends were injectable and had decreased dissociation and degradation in the enzyme lysozyme compared to blends without mannitol. Vancomycin and amikacin eluted in a burst response, with active concentrations extended to seven days compared to five days for blends without mannitol. Mannitol eluted from the paste in a burst the first day and continued through Day 4. Eluates from the mannitol pastes with and without antibiotics decreased viability of established *S. aureus* biofilm by up to 95.5% compared to blends without mannitol, which only decreased biofilm when loaded with antibiotics. Cytocompatibility tests indicated no adverse effects on viability of fibroblasts. In vivo evaluation of inflammatory response revealed mannitol blends scored within the 2–4 range at Week 1 (2.6 ± 1.1) and at Week 4 (3.0 ± 0.8), indicative of moderate inflammation and comparable to non-mannitol pastes (*p* = 0.065). Clinically, this paste could be loaded with clinician-selected antibiotics and used as an adjunctive therapy for musculoskeletal infection prevention and treatment.

## 1. Introduction

Trauma patients, including those involved in car accidents or sports injuries, are at particular risk for developing an infection due to bacterial contamination at the time of injury [1,2]. Traumatic injuries typically require surgical procedures with implanted biomaterials to promote repair, which further increase the risk of infection [2]. Bacteria introduced via environmental exposure, surgical contamination, or hematogenous infection can form biofilm on both tissue and biomaterial implants used for orthopedic fixation or tissue replacement [3,4]. Over $10 billion is spent annually by the United States health care system to treat surgical site infections (SSI), defined as any infection occurring within 30 days following the operation or within a year if an implant was utilized [5,6]. Approximately 2% to 5% of all surgeries performed will result in an SSI amounting to over 300,000 patients per year in the US [7,8]. An SSI involving biofilm in the context of a periprosthetic joint is particularly problematic, resulting in the need for a costly revision surgery, longer hospital stays, lengthened rehabilitation times, and the extended use of antibiotics and analgesics [4,9,10]. 

Biofilm-based infections are difficult to treat due to a number of reasons, including the presence of a subset of bacteria that enter a state of semidormancy with reduced metabolism, known as persister cells [11,12]. Persister cells contribute to an inherent antibiotic resistance allowing biofilm to withstand up to 1000x the therapeutic concentrations of antibiotics [12,13]. In addition to high doses of systemic antibiotics, novel strategies in development to combat biofilm include implant coatings [14], fatty acids [15], matrix-degrading enzymes [16], and quorum-sensing inhibitors [17]. Studies have demonstrated that some bacterial metabolites stimulate the awakening of persisters, allowing antibiotics to more effectively eradicate biofilm [18,19,20]. Mannitol, a sugar polyol naturally found in marine algae and a common metabolite of bacteria, was shown by Allison et al. to shift metabolic reactions within persister cells, stimulating proton motive force and leading to increased uptake of internally acting aminoglycoside antibiotics [21]. The combination of mannitol and gentamicin was shown to decrease biofilm of *Staphylococcus aureus* and *Escherichia coli* both in vitro and in vivo in a murine urinary tract infection model [19]. Recurring infections are likely if persister cells are not eradicated by either tissue debridement during trauma or revision surgery, antibiotics, or the host immune response [22,23,24]. For traumatic injuries, preventative measures such as antibiotic prophylaxis may not be implemented in time or may be insufficient to reduce microbial bioburden in the wound bed [25,26]. With the increased patient morbidity and societal costs of biofilm-associated infection, there is a need to develop an effective strategy capable of targeting persister cells within a biofilm. Currently there are no local delivery systems to codeliver mannitol and antibiotics in the context of musculoskeletal infection. 

Local delivery systems used clinically include nondegradable antibiotic loaded bone cement, calcium sulfate, and polymeric sponges and gels [27,28,29,30,31]. Chitosan, an abundant natural biopolymer derived from the exoskeletons of crustaceans, has been developed into a variety of therapeutic delivery systems due to its beneficial characteristics including: biocompatibility, ability to degrade in vivo, mucoadhesivity, and intrinsic antimicrobial properties [32,33]. Injectable chitosan pastes have been previously evaluated as local antibiotic delivery systems, offering advantages of conforming to complex wound bed geometries [34,35,36,37]. Chitosan pastes and sponges have been shown to deliver aminoglycoside and glycopeptide antibiotics at concentrations high enough to eradicate biofilm locally to the wound bed [38,39,40,41], but remaining persister cells could lead to recurrent infection. In this study, we fabricated biodegradable and injectable blends of mannitol, chitosan, and polyethylene glycol (PEG) to evaluate the hypothesis that elution of both antibiotics and mannitol enhances eradication of biofilm-associated *S. aureus*. Elution, antimicrobial efficacy, and biocompatibility were evaluated using in vitro and in vivo preclinical models.

## 2. Results

### 2.1. Material Characteristics

#### 2.1.1. Assessment of Injectability 

During the injectability test performed with the 1 mL syringe (no needle), no significant difference was detected between chitosan paste with mannitol (ChMPEG) and without mannitol (Ch-PEG) with average ejection forces of 1.20 N ± 0.20 N and 1.82 N ± 0.38 N, respectively (*p* = 0.18) (Table 1). 

#### 2.1.2. Enzymatic Degradation

The ChMPEG paste continued to degrade over the 14-day period, only degrading by 28.2 ± 7.7%, while the chitosan paste without mannitol reached a plateau after the first day and remained at 50.0 ± 3.2% through Day 14 (Figure 1). For each time point, ChMPEG paste had significantly higher amount of material remaining (*p* < 0.001). 

### 2.2. Elution Profiles

#### 2.2.1. Antibiotic Elution: Vancomycin and Amikacin

Over the course of the seven-day elution study, both groups released both amikacin and vancomycin in a burst, but Ch-PEG released approximately three times the amount of antibiotics compared to ChMPEG on the first day (*p* < 0.001) (Figure 2a,b). The ChMPEG paste eluted both vancomycin and amikacin at detectable levels for seven days, with levels above MIC sustained for seven days for vancomycin and five days for amikacin (Figure 2).

#### 2.2.2. Mannitol Elution from ChMPEG Paste 

Mannitol eluted in a similar burst profile with 696 ± 35 µg/mL released the first day, and no detectable release after Day 4 (Figure 3). 

### 2.3. Antibiotic Activity Against Staphylococcus Aureus

The results of the zone of inhibition (ZOI) study against *S. aureus* (Figure 4 and Table 2) indicate the ChMPEG paste was able to produce measurable zones from Day 1 (5.31 ± 0.19 mm) until Day 7 (0.22 ± 0.04 mm), while the Ch-PEG paste only had measurable zones from Day 1 (6.18 ± 0.16 mm) until Day 4 (0.42 ± 0.19 mm). 

### 2.4. Biofilm Eradication

For the paste eluate study against established *S. aureus* biofilm on MBEC™ peg plates, all groups except Ch-PEG with no antibiotics showed a decrease in bacterial viability (Figure 5). Blends of paste containing mannitol in addition to antibiotics showed a decrease in bacterial viability normalized to control wells with no chitosan paste added, although no statistical difference was detected between paste blends with antibiotics. 

### 2.5. Cytocompatibility & Biocompatibility

#### 2.5.1. In Vitro Cytocompatibility with NIH-3T3 Cells

The percent cell viability of fibroblasts exposed to both Ch-PEG and ChMPEG pastes exceeded 100% and were not statistically different when compared to untreated controls (Figure 6). Both were also above the accepted 70% value when normalized to the cells with no treatment standard, in accordance with the ISO 109935 Biological Evaluations of Medical Devices standard when evaluating biomaterials against fibroblasts. 

#### 2.5.2. Biocompatibility Model with Sprague Dawley Rats

Representative histological sections of each paste group showed that paste groups elicited mild to moderate inflammation in the rat biocompatibility model with evidence of inflammatory cells at the periphery of implant material and migrating into the chitosan pastes (Figure 7a). Histological scores by blinded raters (n = 5) indicated no significant difference between the inflammatory responses of the two paste groups (Figure 7b). 

## 3. Discussion

The mannitol–chitosan blend was able to form an injectable biomaterial capable of being loaded at the time of care with clinician-selected antibiotics. This material shows potential to serve as a degradable local antibiotic delivery system after surgery or traumatic injury, and could be particularly useful during revision surgeries following periprosthetic joint infections. When used in conjunction with systemic or locally delivered antibiotics, the added layer of antimicrobial protection provided by this delivery system could provide additional coverage against strains that are not susceptible to systemic or local antibiotics administered or against organisms that have formed biofilm on the surface of implants or nonviable tissue. This in turn could decrease the number of SSI and recurring infections following traumatic injuries and thus also decrease health care costs as well as patient hospital stays and rehabilitation times.

The extended degradation and elution of antibiotics from mannitol blends may be due to the potential formation of a polyelectrolyte complex through hydrogen-bonding interactions between mannitol hydroxyl groups and the hydroxyl groups present on chitosan, a hypothesis supported by previous studies [42,43]. The most commonly reported polyelectrolyte complex between chitosan and polyols is the thermogelling combination of beta-glycerophosphate and chitosan [36,44,45]. One beta-glycerophosphate/chitosan local delivery system reported by Boles et al. only eluted amikacin and vancomycin until Day 5, with antimicrobial activity against *S. aureus* for only three days [36]. The increased duration of elution for the mannitol blend could be attributed to differences in the amount of aqueous solution used for hydration as well as the antibiotic loading; less water within the system could lead to stronger intermolecular interactions between chitosan, extending the delivery even when the total amount of antibiotic loaded is lower [35,36]. In chitosan pastes with PEG only, both the elution of antibiotics and antimicrobial activity were only previously reported for 72 h [34,35]. The burst release of mannitol and cessation of release after Day 4 also support that a polyelectrolyte complex may form, with mannitol interactions with hydroxyl groups of chitosan limiting duration of release compared to antibiotics. The properties of chitosan biomaterials can also be tailored by the degree of deacetylation, molecular weight, temperature, and pH [46,47,48,49]. While these parameters were not assessed during this investigation, it may be possible to tailor the release characteristics and other properties through the control of these factors. 

The mannitol component of the mannitol–chitosan paste may combat biofilm formation by shifting metabolic reactions within the persister cell phenotype. This activation of persister metabolism could increase the uptake of aminoglycosides delivered by the paste and therefore increase the overall antimicrobial activity [19,20]. However, the ChMPEG paste eluates exhibited antibiofilm properties even in the absence of antibiotics when evaluated against an established *S. aureus* biofilm. Mannitol’s effect on persister cell metabolic reactions may also increase susceptibility to both the acetic acid solvent of the paste and the cationic chitosan polymer. Chitosan has been shown to have inherent antimicrobial properties and exhibit antimicrobial activity against *E. coli*, *Pseudomonas aeruginosa, Staphylococcus epidermidis*, and *S. aureus* [50]. The actual mechanism of chitosan’s antimicrobial activity is not yet fully understood but may be attributed to chitosan’s intrinsic cationic nature, low molecular weight, and its ability to penetrate bacterial cell walls, bind with DNA, and inhibit transcription [51,52,53]. Mannitol combined with chitosan may allow for increased uptake of chitosan, thereby increasing the antimicrobial activity of chitosan within persister cells. When used in conjunction with systemically or locally delivered antibiotics, this biofilm eradication activity of the material itself could provide an additional level of protection against implant-associated infection due to biofilm.

Cytocompatibility studies show similar results to the chitosan injectable pastes described by Berretta et al., and indicate that mannitol’s addition to the system does not affect the cellular response to the paste [34]. Mannitol is a Food and Drug Administration (FDA)-approved food sweetener, as well as a medication used as an osmotic agent to decrease swelling in the brain, to prevent acute renal failure, and to treat pulmonary conditions like cystic fibrosis by drawing fluid out of the lungs [54,55]. For these reasons, no adverse side effects were expected to be observed upon the addition of mannitol to the system [56]. Chitosan is also known to be biocompatible and degrades within the body through lysozyme degradation of its polymer chains and subsequent dissociation of these polymer chains in acidic conditions [57,58]. Chitosan first breaks down into long oligosaccharide chains and continues to degrade until reaching its final end product of glucosamine [59], a natural component of tissues that promotes healing [60]. The acidic dissolution of chitosan and ensuing inflammation could be higher in the form of a hydrated paste, which may explain the increased inflammation seen in vivo compared to neutralized chitosan sponges [61]. This would cause a greater inflammatory response than previously reported beta-glycerophosphate chitosan paste, which was attributed to the neutralizing effect of beta-gylcerophosphate [36]. Another possible explanation for the inflammatory response seen with the ChMPEG paste is the potential for mannitol to act as an osmotic diuretic. The osmotic effect of mannitol when introduced into the body is confined to the extracellular space and can draw fluid from the intracellular space of the surrounding cells to maintain osmotic equilibrium [56]. Active inflammation response with macrophage and neutrophil presence due to the innate immune response may contribute to a microenvironment unsuitable for biofilm growth or phagocytosis of pathogenic microorganisms. 

In conclusion, the mannitol–chitosan blend was capable of serving as an injectable biomaterial demonstrating an improved elution profile, enhanced antimicrobial properties, and antibiofilm properties observed with and without antibiotics, in addition to exhibiting cytocompatibility and biocompatibility. This material could be clinically applied in the context of an injectable, infection-prevention paste either loaded with antibiotics or as a stand-alone prophylactic measure. Such injectable materials have advantages in conforming to complex geometries of the wound bed and in the ability to customize the types of antimicrobials loaded. Future studies will include the evaluation of the ChMPEG paste’s efficacy at eradicating established biofilm in vivo. Subsequent in vitro tests will measure the antibiofilm properties of the mannitol–chitosan blend against different species of bacteria relevant to biofilm-associated infections including methicillin-resistant *S. aureus*, *S. epidermidis*, and *E. coli*, as well as *P. aeruginosa*. Additionally, in vivo models to evaluate the paste’s efficacy for the prevention of infection both with and without adjunct antibiotics will need to be evaluated. Future studies will use fluorescent microscopy to further evaluate biofilm effects and characteristics. Finally, due to the complex inflammatory response during infection, future exploration of the efficacy of chitosan–mannitol blends will be conducted to elucidate the effect of this biomaterial on the body’s response to infection. 

## 4. Materials and Methods 

### 4.1. Fabrication

Chitopharm S chitosan powder (Chitinor AS, Tromsø, Norway) (82.46 ± 1.679 degree of deacetylation and 250.6 kilodalton average molecular weight) at 1% (*w*/*v*) and 8,000 g/mol PEG (Sigma Aldrich, St. Louis, MO, USA) at 1% (*w*/*v*) were dissolved in 0.85% acetic acid in deionized water solution to form the control paste (Ch-PEG). Additionally, 2% mannitol (*w*/*v*) (Bulksupplements.com) was dissolved in the previous solution to form the mannitol paste group (ChMPEG). The solutions were cast in 25 mL aluminum dishes and frozen overnight at −80 °C, then lyophilized in a benchtop freeze dryer (LabConco, Kansas City, MO, USA) to create acidic dehydrated sponges. The sponge types were then ground separately into a fine powder and stored in a desiccator until use. Preliminary determination of the hydration ratio was achieved by evaluating consistency of the paste when mixed by coupling (Interlok™, Qosina) a 3 mL syringe filled with 1.25 mL of phosphate-buffered saline (PBS) to a 10 mL syringe with 500 mg of ground powder and injecting back and forth between syringes until evenly mixed.

### 4.2. Material Characteristics

#### 4.2.1. Assessment of Injectability through 1 mL Syringe

Using an Instron Universal Testing Machine with a 5 kN load cell force plate for each paste composite, an injectability test was performed through a 1 mL syringe (BD Products). The force required to eject the paste (n = 3) was automated by Instron’s Bluehill 2 Software while compressing the syringe plunger at 1 mm/sec for a specified length, 50 mm. The 1 mL syringe was loaded with 1.0 mL of paste and the maximum force detected was used for comparison between paste groups. 

#### 4.2.2. Enzymatic Degradation of Paste

Amounts of 0.5 grams of each paste type, with different samples used for each time point, were hydrated with PBS and approximately 0.3 mL (n = 3) of hydrated paste was placed in a 5 mL working volume petri dish (Nunclon). The degradation solution was prepared by dissolving 1 mg/mL lysozyme type VI (MP Biomedicals) and 100 µg/mL Normocin antibiotic/antimycotic in PBS. Next, 5 mL of degradation solution was added to the petri dish and samples were placed in a 37 °C incubator. Degradation solution was refreshed every other day by aspirating the media and adding 5 mL of fresh solution. Samples were collected at the following time points: 1, 3, 5, 7, and 14 days. Degradation solution was siphoned off and the samples were placed in an oven at 45 °C. After 24 h of drying, samples were weighed and weighed again after another 24 h until samples reached a constant mass, before comparing their final weight to their initial weight to determine the degradation rate.

### 4.3. Elution Profiles 

Antibiotics were loaded into the paste by mixing with a solution of vancomycin and amikacin, both at a 10 mg/mL concentration, in PBS. Pastes (0.3 mL, n = 3) were injected into a 12-well CellCrown^TM^ insert fitted with a 44 µm pore size nylon filter and submerged in 4 mL of PBS. Sampling occurred daily for seven days with addition of fresh PBS after each sample was taken. Antibiotic concentrations were evaluated with high-performance liquid chromatography (HPLC) using a ThermoScientific Dionex Ultimate 3000 Series HPLC system and a BDS Hypersil reversed-phase C18 column (150 × 4.6 mm) at 30 °C. Vancomycin was detected with a UV/Vis spectrophotometer at 209 nm in a mobile phase of 20% acetonitrile and 80% phosphate buffer at pH 7.4 [62]. Amikacin was quantified using precolumn derivatization with an o-phthaldialdehyde reagent and subsequent detection using a fluorescence detector [63]. Briefly, 50 mg derivatization agent of o-phthaldialdehyde in 0.5 mL methanol and 100 µL mercaptoethanol in water was adjusted to pH 10 using sodium hydroxide. Using a programmed routine, the autosampler needle withdrew 1 µL of sample, 5 µL of derivatization reagent, and 2 µL acetic acid. This mixture was allowed to react for 60 s and injected into a BDS Hypersil reverse phase C18 column in a mobile phase of 30% acetonitrile and 70% phosphate buffer at pH 7.4. Excitation was 340 nm and emission was monitored at 455 nm [63]. Mannitol was detected using a charged aerosol detector (Corona VEO, Thermoscientific, Waltham, MA, USA) using a hydrophilic interaction liquid chromatography (HILIC) column (ThermoFisher Scientific, Waltham, MA, USA) using an isocratic solvent consisting of 80% acetonitrile and 20% water. 

### 4.4. Antibiotic Activity

Activity of antibiotics in eluate samples (n = 3) was assessed using ZOI assays. Bacterial lawns were grown on trypticase soy agar plates by inoculating 100 µL of a 10^6^ colony-forming unit (CFU) concentration of *S. aureus* (UAMS-1) onto each plate. Eluate samples from days 1–7 (30 µL) were loaded onto sterile paper discs (6 mm = diameter, BD Products) and placed on the bacterial lawns and allowed to incubate at 37 °C for 24 h. The plates were then photographed and ZOI diameters were measured using ImageJ Software (NIH) (version 1.52e, Bethesda, MD, USA). Positive controls included a disc at the center of each plate loaded with 30 µg/mL of antibiotics, as well as standard antibiotic concentrations decreased 0.5-fold from 5.0 mg/mL to 0.004 mg/mL. Any zone measuring below 0.2 mm was reported as 0 mm in Table 2. 

### 4.5. Biofilm Eradication

Antimicrobial activity of each paste type’s eluates was evaluated against established biofilms of *S. aureus* (UAMS-1) grown in MBEC™ 96-well plates with 150 µL at concentrations of 10^6^ CFU in trypticase soy broth (TSB). Biofilm was grown for 24 h at 37 °C on an orbital shaker at a speed of 220 rpm (MIDSCI™ LabDoctor Horizontal Shaker). The pegs were submerged in 150 µL (n = 6) of the first-day eluate sample taken from the elution study (Day 1 vancomycin concentration = 0.18 mg/mL for ChMPEG and 0.55 mg/mL for Ch-PEG; Day 1 amikacin concentrations = 0.30 mg/mL for ChMPEG and 0.67 mg/mL for Ch-PEG). Controls of 0, 0.05, and 5 mg/mL of vancomycin and amikacin in PBS were also included as negative and positive controls. After 24 h, the tops of the MBEC™ plates were removed, placed in fresh TSB, sonicated for 5 min at 40 kHz (Fisher Scientific Ultrasonic Bath, 9.5 L), and incubated for 24 h. PrestoBlue™ viability reagent was used to compare bacterial survival on pegs and growth after exposure to the eluates. PrestoBlue™ reagent was added to each well according to the manufacturer’s instructions, and plates were incubated at 37 °C for 1 h. Fluorescence was determined with an excitation value of 540 nm and an emission value of 570 nm using a Biotek Synergy™ H1 microplate reader, with increased fluorescence indicating a higher number of viable cells. Percent viability was determined using the control with PBS only (0 mg/mL antibiotic). 

### 4.6. Cytocompatibility 

NIH 3T3 fibroblasts (ATCC® CRL-1658™) were seeded at 10^4^ cells/cm^2^ in a 24-well plate and grown in Dulbecco’s Modified Eagle’s Medium supplemented with 10% fetal bovine serum and 100 µg/mL of Normocin antibiotic/antimitotic solution for 24 h at 37 °C and 5% CO_2_. The different paste types were compared to cells not exposed to chitosan pastes. Samples were sterilized with ethylene oxide gas (EtO) prior to testing. The pastes were hydrated with PBS and a volume of 0.2 mL was inserted into cell culture inserts (Falcon, pore size = 8 µm). Cell viability (n = 3) was quantified using CellTiter-Glo® (Promega) after 24 h of exposure to the pastes. CellTiter-Glo® reagent was added to each well at a ratio of 1:2 reagent to cell culture media and incubated for 10 min. After incubation, 125 µL was transferred to an opaque 96-well plate and luminescence was measured with a Biotek Synergy H1 microplate reader. Percent viability was calculated as the luminescence of the sample compared with the mean value for wells with cells not exposed to chitosan pastes. 

### 4.7. Biocompatibility

This animal model was approved by the Institutional Animal Care and Use Committee (IACUC) at the University of Memphis (#0811). Sixteen male Sprague Dawley rats (~375 g) were divided into two groups, ChMPEG paste and Ch-PEG paste, and two time points (n = 8/group/time point). Rats were anesthetized using isoflurane inhalation. Dorsa were shaved and cleaned with betadine and isopropanol. Each rat received all implant groups, inserted at separate and randomized dorsal locations with the midline separating the groups. After approximately 1 cm long incisions were made and surgical scissors were used to create the subcutaneous pouch, 0.3 mL of pastes hydrated with PBS were placed subcutaneously. At the two time points of 1 week and 4 weeks, rats were sacrificed and the skin tissue surrounding the remaining material was harvested and preserved in a 10% neutral formalin solution for a minimum of 14 days before histological processing. Tissue samples embedded in paraffin, cut into 5 µm thick sections, and stained with hematoxylin and eosin (H&E) for histological analysis. Reviewers blinded to group identification (n = 5) rated inflammation on a scale of zero to five, based on the scale determined by Jansen et al., wherein zero indicates no inflammation and five indicates severe inflammation [64]. 

### 4.8. Statistical Analysis

Statistical analysis was performed using SigmaPlot 14 (Systat Software Inc., San Jose, CA, USA) and GraphPad Prism 7.2 software (GraphPad Software Incorporation, La Jolla, CA, USA). Data is reported as mean ± standard deviation. The elution data was assessed with a two-way analysis of variance (ANOVA) followed by Holm–Sidak post hoc analysis to detect significant differences with time and experimental groups. Eradication results were assessed using Kruskal–Wallis one-way ANOVA followed by a Tukey post hoc test. Cytocompatibility results were assessed with a one-way ANOVA followed by Holm–Sidak post hoc analysis used to detect statistical differences among experimental groups. In vivo functional compatibility results were nonparametric and therefore analyzed using a Kruskal–Wallis one-way ANOVA followed by a Tukey post hoc test. Results were considered statistically significant when *p* < 0.05. 

## Figures and Tables

**Figure 1 marinedrugs-17-00517-f001:**
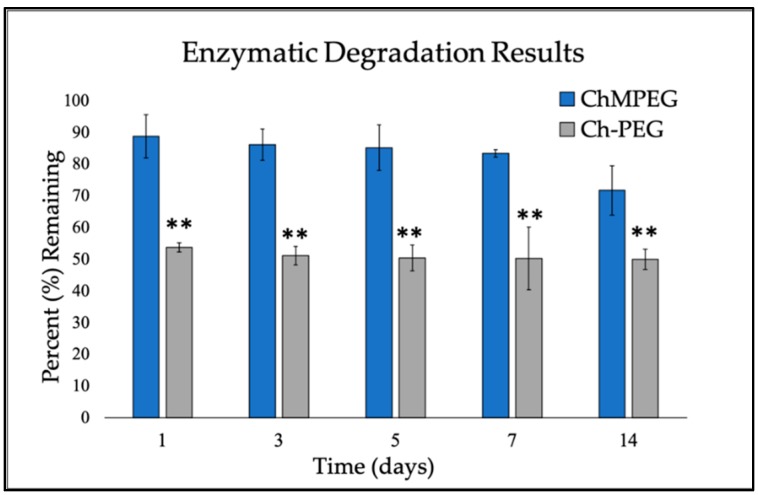
Enzymatic degradation of paste. Graph represents percent of original paste mass remaining during enzymatic degradation of the Ch-PEG and ChMPEG pastes with a 1 mg/mL lysozyme solution over a 14-day period for each time point. Each bar represents mean (n = 3) and error bars indicate standard deviation. ** indicates significant difference between groups detected using two-way ANOVA with Holm–Sidak post hoc tests (*p* < 0.001).

**Figure 2 marinedrugs-17-00517-f002:**
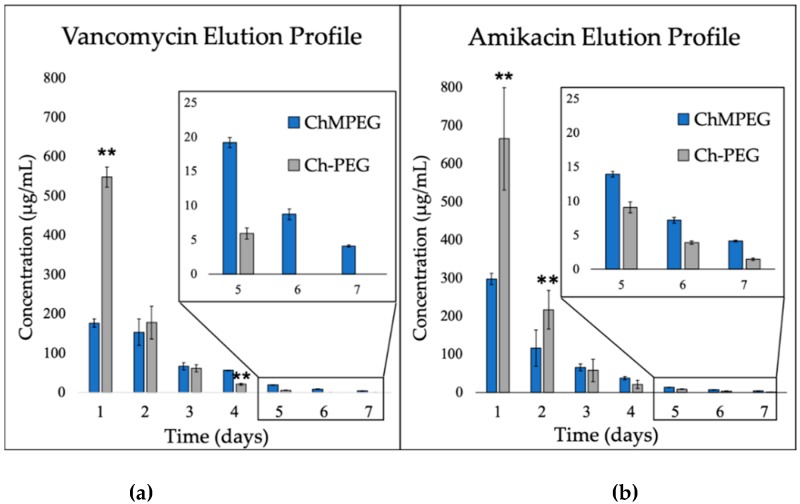
Vancomycin and amikacin elution profiles. Graphical representations show the mean elution concentrations of (**a**) vancomycin and (**b**) amikacin detected for each paste group (n = 3). Inset graphs illustrate the elution data for days 5–7 for both vancomycin and amikacin. Error bars indicate standard deviation. ** indicates significant difference between groups detected using two-way ANOVA with Holm–Sidak post hoc tests (*p* < 0.001).

**Figure 3 marinedrugs-17-00517-f003:**
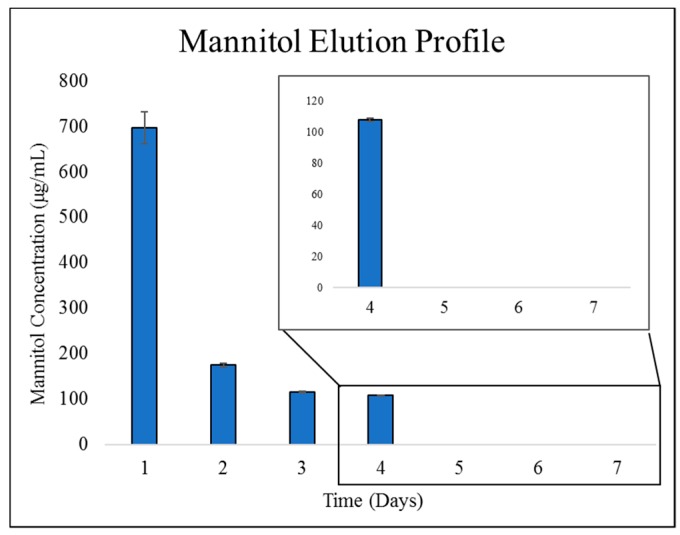
Mannitol elution profile. Graph shows the mean mannitol concentrations eluted from the ChMPEG paste (n = 3). Inset graphs illustrate elution data for days 4–7. Error bars indicate the standard deviation.

**Figure 4 marinedrugs-17-00517-f004:**
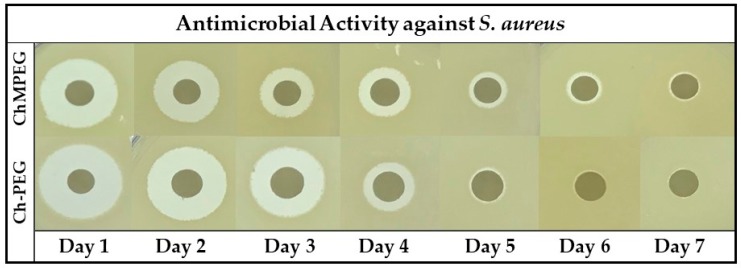
Antimicrobial activity against *S. aureus*. Figure illustrates representative images of the ZOI against *S. aureus* created around the 6 mm paper discs loaded with different paste eluate samples from Day 1 through Day 7.

**Figure 5 marinedrugs-17-00517-f005:**
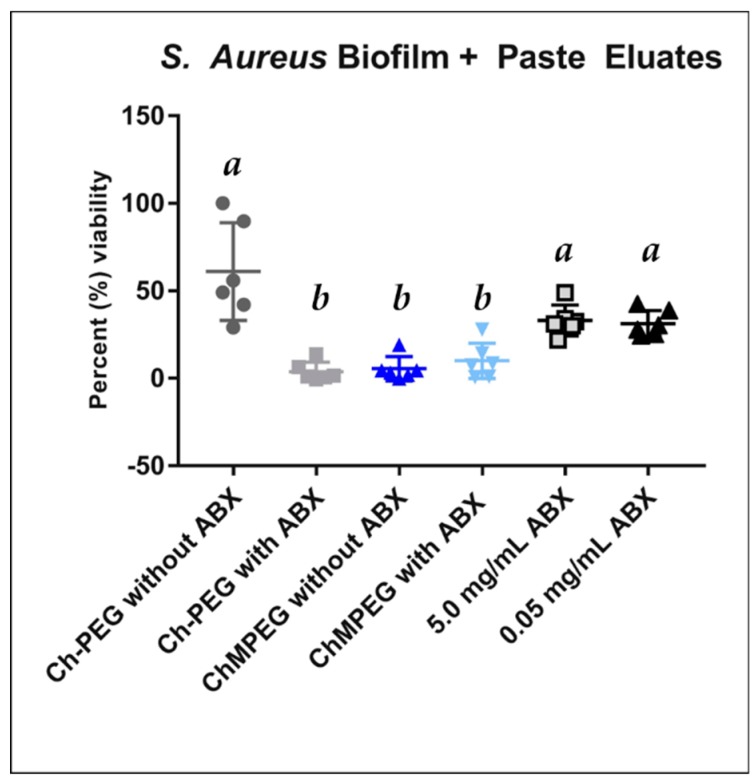
*S. aureus* biofilm + paste eluates. Scatterplot represents bacterial viability of *S. aureus* after direct contact with Day 1 eluates from pastes (n = 6) and two standard antibiotic concentrations. Percent bacterial viability was determined by comparison to viability of nontreated controls. The abbreviation ABX indicates antibiotics, which were a mix of vancomycin and amikacin. Groups with the same letters above indicate that groups are not significantly different when tested with one-way Kruskal–Wallis ANOVA and Tukey post hoc tests (*p* > 0.05).

**Figure 6 marinedrugs-17-00517-f006:**
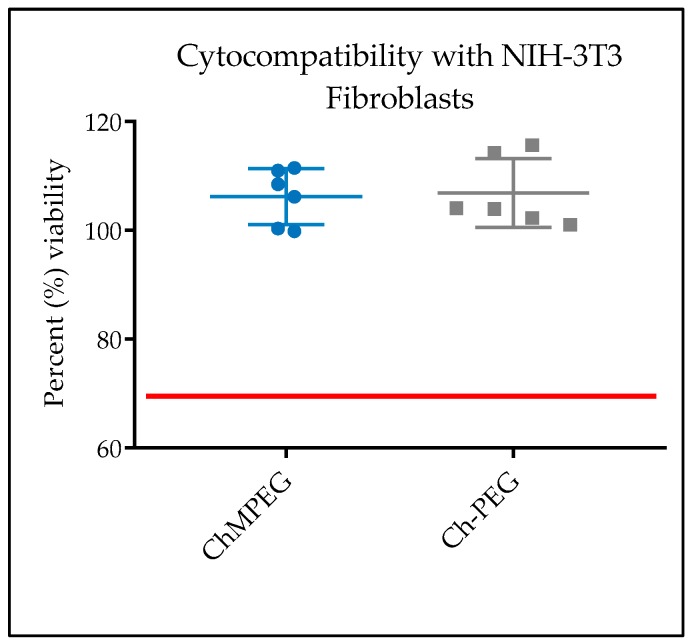
Paste cytocompatibility assessed with NIH-3T3 fibroblasts. Graphical representation of percent NIH-3T3 cell viability for each paste type determined as compared to cell viability in untreated control wells. Error bars indicate standard deviation and the red line represents the accepted lower threshold value of 70% according to ISO 109935. There were no statistically significant differences between groups or between groups and untreated controls.

**Figure 7 marinedrugs-17-00517-f007:**
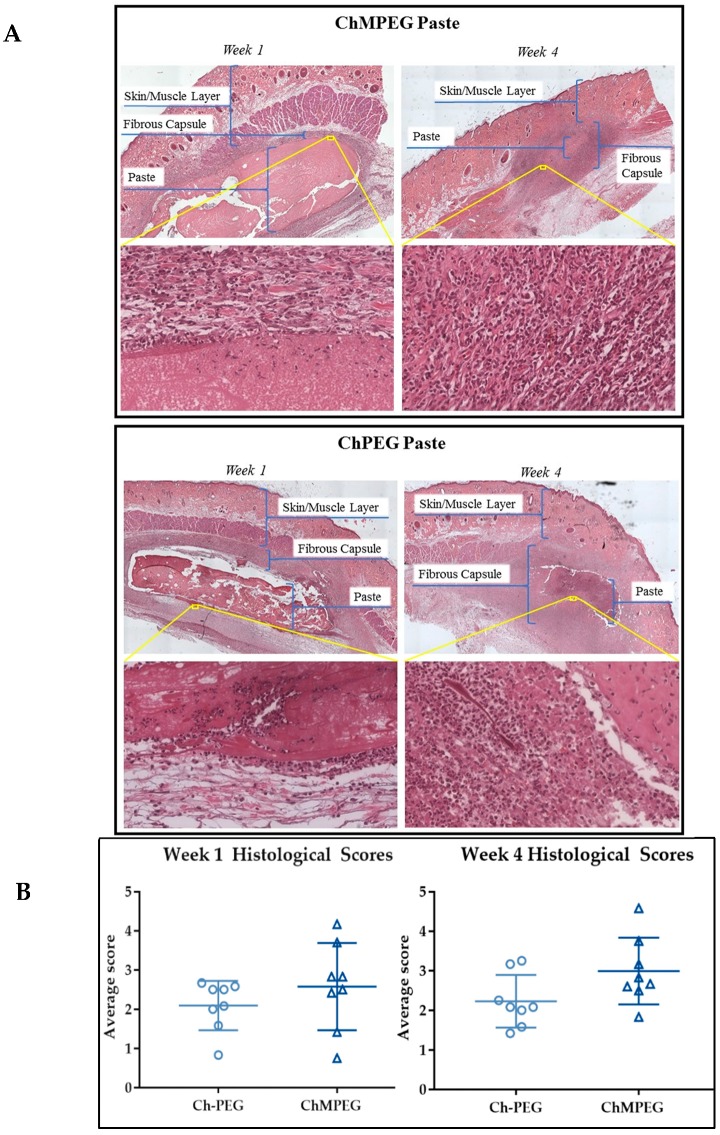
(**A**) ChMPEG and Ch-PEG paste histological sections. Comparison of Week 1 and Week 4 inflammatory response with two representative images of 5 μm sections stained with H&E for subcutaneous tissue injected with ChMPEG (**top**) and Ch-PEG paste (**bottom**) (n = 8). (**B**) Week 1 and Week 4 histological scores. Scatter plots show mean histological scores for each rat (n = 8) in biocompatibility model. No significant differences were seen between groups (*p* > 0.05).

**Table 1 marinedrugs-17-00517-t001:** Average force required to eject paste. Values reported are the mean force ± standard deviation needed to eject each paste group from a 1 mL syringe (n = 3). No statistical differences were detected using one-way ANOVA.

*Paste Group*	Force (N)
*ChMPEG*	1.20 N ± 0.20 N
*Ch-PEG (control)*	1.82 N ± 0.38 N

**Table 2 marinedrugs-17-00517-t002:** ZOI against *S. aureus*. Average ZOI measured with ImageJ (NIH) for each paste type for each eluate sample day (n = 9) against *S. aureus*. The ± indicates standard deviation. The * indicates the groups were significantly different from one another on the corresponding day (*p* < 0.001 for Day 1–6 and *p* = 0.031 for Day 7) detected using a two-way ANOVA with Holms–Sidak post hoc tests.

	ChMPEG ZOI	Ch-PEG ZOI
*Day 1*	5.31 ± 0.19 mm	6.18 ± 0.16 mm *
*Day 2*	4.37 ± 0.31 mm	4.92 ± 0.29 mm *
*Day 3*	2.92 ± 0.19 mm	1.99 ± 0.08 mm *
*Day 4*	2.40 ± 0.20 mm	0.42 ± 0.19 mm *
*Day 5*	1.18 ± 0.17 mm	0 mm *
*Day 6*	0.61 ± 0.32 mm	0 mm *
*Day 7*	0.22 ± 0.04 mm	0 mm *

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
