# Peer review of "Characterization and Antibiofilm Activity of Mannitol–Chitosan-Blended Paste for Local Antibiotic Delivery System"

_marinedrugs, 2019, doi:10.3390/md17090517_

Round 1

Reviewer 1 Report

In ther manuscript submitted by Jennings et al. authors describe the utility of mannitol-chitosan blend as "a degradable local antibiotic 156 delivery system after surgical incision or traumatic injury and could also be particularly useful during 157 revision surgeries following periprosthetic join infections"

The manuscript is well written and the reported results are interesting since biofilm formation is an important virulence factor which is responsible of severe chronic infectionsparticularly relevant in implnts of prostheses, catheters and tissue replacements.

In the introduction recent references concerning on the abtibiotic resistance related to bacterial biofilm formation should be added (e.g. European Journal of Medicinal Chemistry, 2019, 161, pp. 154-178 and Journal of Medicinal Chemistry 2017

60(20), pp. 8268-8297 ).

All the results must be clarify with  more detailed description of the obtained data.

All figures must be commented and explained in the text.

After these modifications will be made, ther manuscript can be accepted for the pubblication.

Author Response

Review 1

Comments and Suggestions for Authors

In their manuscript submitted by Jennings et al. authors describe the utility of mannitol-chitosan blend as "a degradable local antibiotic 156 delivery system after surgical incision or traumatic injury and could also be particularly useful during 157 revision surgeries following periprosthetic join infections"

The manuscript is well written and the reported results are interesting since biofilm formation is an important virulence factor which is responsible of severe chronic infections particularly relevant in implants of prostheses, catheters and tissue replacements.

Thank you for your time and for your helpful comments. Below are specific areas of the manuscript that have been changed.

In the introduction recent references concerning on the antibiotic resistance related to bacterial biofilm formation should be added (e.g. European Journal of Medicinal Chemistry, 2019, 161, pp. 154-178 and Journal of Medicinal Chemistry 2017 60(20), pp. 8268-8297).

We have added these references as well as others to better report the most recent biofilm research.

All the results must be clarify with more detailed description of the obtained data.

Results section was updated to include specific values reported in the figures and tables.

All figures must be commented and explained in the text.

We have checked and rechecked that all figures are referenced in the text, but if there is something specific that was not referenced, we will edit.

After these modifications will be made, their manuscript can be accepted for the publication.

Reviewer 2 Report

In the MS entitled “Injectable mannitol chitosan blended past eradicates S. aureus biofilms and elutes antimicrobials.” Pace et al has shown the role of Ch-PEG and ChMPEG as a biodegradable adjuvant therapy for musculoskeletal infection prevention for Surgical Site Infection (SSI). I have many concerns with this MS. The MS is at quite an immature stage, the MS is poorly written and lacks basic information as if how the experiments are done.

Abstract:

What is Moderate inflammatory response? Authors should mention how they came to this conclusion (p-value?).

At the end of the abstract authors claims that mannitol paste with or without antibiotics decreases bacterial viability, I am surprised by this conclusion, do authors claim that there is no role of antibiotics over S. aureus and mannitol paste can take place of antibiotics. The statement should be rewritten or explained properly.

Introduction:

Lack of literature cited, there is lots of literature about the role of metabolites and biofilm dispersion. Authors need to cite the latest known facts about this important aspect of biofilm research.

Results:

Results are encouraging but the not concluded in a scientific way, I strongly believe that the paper should be retitled or more emphasis should be given on S. aureus biofilms or mixed biofilms. Authors are trying to prove that Ch-PEG is better than ChMPEG but not through the activity against biofilm but by their enzymatic degradation.

I  do not understand the concept of using just syringe and syringe with 18 G needle, what is the takeaway message, how authors plan to inject paste in a patient just with the syringe. It’s confusing and needs to be addressed.

Figure 1 – Figure indicates Ch-PEG is more degradable then CHMPEG, where is the statistical data? No p-value? How the conclusion was reached?

Figure 2 – Same problem in antibiotic and mannitol elution. It’s not clear from the Materials and Methods sections how the conclusion was made.

Figure 4 – I personally cannot see any difference in the antimicrobial activity against S. aureus, why none of the figures have statistical data?

Figure 5 – Data indicates that there is not any role of antibiotics in biofilm eradication. Though antibiotic, it shows almost 40% of bacterial eradication. How this was tested? Plate assay? Viability assay?

Figure 7 – Biocompatibility assay – The magnification of the H&E image is too low to see if there is any kind of immune response, in forms of granulocytes. Authors must zoom in to see if there is a migration of neutrophils.

How thick is the section?

Figure 8 should be Figure 7 (b) and no statistical data shown for the histological scores.

It will be helpful for the readers of the MS, if more details about how the MS is grown and how the eradication studies were done, some fluorescent microscopy pictures will help to understand the impact of eradication.

For the scientific point of view, it will be very interesting to know what was the immunological response of the eradication of the biofilms.

Author Response

Review 2:

Comments and Suggestions for Authors

In the MS entitled “Injectable mannitol chitosan blended past eradicates S. aureus biofilms and elutes antimicrobials.” Pace et al has shown the role of Ch-PEG and ChMPEG as a biodegradable adjuvant therapy for musculoskeletal infection prevention for Surgical Site Infection (SSI). I have many concerns with this MS. The MS is at quite an immature stage, the MS is poorly written and lacks basic information as if how the experiments are done.

Thank you for your time and for your helpful comments. This is an ongoing study reporting material characteristics determined by in vitro and in vivo evaluations for this new paste material.  Future and ongoing expansion includes additional in vivo testing, including in infected models. Would you suggest that we add “exploratory” into the introduction or the title to clarify that this text reports preliminary results? The description of all methods have been updated to include more detail about how the experiments were conducted.    

Abstract:

What is moderate inflammatory response? Authors should mention how they came to this conclusion (p-value?).

The text was updated to include more detail about the moderate inflammatory response conclusion based on a 5 point histological scale. The p value was added to this sentence of the abstract. We cannot reference the study we used to determine inflammatory response rankings within the abstract, but it has been referenced in the methods.

At the end of the abstract authors claims that mannitol paste with or without antibiotics decreases bacterial viability, I am surprised by this conclusion, do authors claim that there is no role of antibiotics over S. aureus and mannitol paste can take place of antibiotics. The statement should be rewritten or explained properly.

To be clear, we did not develop this mannitol paste to replace antibiotics; however, the observed activity is promising for infections that are not covered by some of the common antibiotics.  The intent of this paste was specifically to be used in conjunction with aminoglycosides to take advantage of the effect reported by Allison et al.  This local delivery system is intended as an adjunct therapy for patients with trauma or surgical procedures who would likely be on systemic antibiotic.  

Introduction:

Lack of literature cited, there is lots of literature about the role of metabolites and biofilm dispersion. Authors need to cite the latest known facts about this important aspect of biofilm research.

Citations in the introduction have been updated to reflect the most recent advances in biofilm research.

Results:

Results are encouraging but the not concluded in a scientific way, I strongly believe that the paper should be retitled or more emphasis should be given on S. aureus biofilms or mixed biofilms. Authors are trying to prove that Ch-PEG is better than ChMPEG but not through the activity against biofilm but by their enzymatic degradation.

We chose to include S. aureus in the title because we only report results from that strain. To clarify about degradation, we are reporting the enzymatic degradation of the material, not enzymatic degradation of biofilm. Furthermore, we are not concluding that the Ch-PEG paste is “better”, but simply reporting that one material degrades in the body faster. Could you provide clarification on how the title should change?  One alternative we considered is “Injectable mannitol-chitosan blended paste local antibiotic delivery system: material characterization and anti-biofilm activity”.

I do not understand the concept of using just syringe and syringe with 18 G needle, what is the takeaway message, how authors plan to inject paste in a patient just with the syringe. It’s confusing and needs to be addressed.

Eventually, we would like a delivery system that is applicable in complex wounds or non-surgical delivery in minimally invasive manner, which is why we investigated its injectable potential through a syringe. We removed the results for the 18G needle, as it does not support this application.

Figure 1 – Figure indicates Ch-PEG is more degradable then CHMPEG, where is the statistical data? No p-value? How the conclusion was reached?

P value and asterisks were added.

Figure 2 – Same problem in antibiotic and mannitol elution. It’s not clear from the Materials and Methods sections how the conclusion was made.

Statistical values were added to these graphs as well.

Figure 4 – I personally cannot see any difference in the antimicrobial activity against S. aureus, why none of the figures have statistical data?

Statistical values were added to Table 2, which numerically describes the zones shown in the images in Figure 4. It may be difficult to see the zones for the later time points in the images, but we did measure zones through the ImageJ software.

Figure 5 – Data indicates that there is not any role of antibiotics in biofilm eradication. Though antibiotic, it shows almost 40% of bacterial eradication. How this was tested? Plate assay? Viability assay?

We are not claiming that antibiotics have no role—they are certainly capable of eradicating biofilm at very high concentrations. A delivery system is capable of achieving these high concentrations locally and would be used as an adjunctive therapy. This test was to evaluate whether the paste materials were able to elute enough mannitol in combination with antibiotics to eradicate biofilm.  It was unexpected that non-antibiotic loaded groups reduced viability to the degree that they did. As stated in the methods section, this was a viability assay using Prestoblue to determine bacteria viability.

Figure 7 – Biocompatibility assay – The magnification of the H&E image is too low to see if there is any kind of immune response, in forms of granulocytes. Authors must zoom in to see if there is a migration of neutrophils.

We have added images at 20x magnification for representative areas at the periphery of the material, which shows granulocytes near and migrating into the implant. 

How thick is the section?

The section thickness is 5 μm. The text has been updated to include this value.

Figure 8 should be Figure 7 (b) and no statistical data shown for the histological scores.

Figure 8 was changed to figure 7b. Statistical data was added to indicate no significant difference between groups (p>0.05).

It will be helpful for the readers of the MS, if more details about how the MS is grown and how the eradication studies were done, some fluorescent microscopy pictures will help to understand the impact of eradication.

We agree that fluorescent images are valuable for illustrating bacterial biofilm characteristics. However, we did not have access to a microscope with high enough magnification and resolution to use this as a confirmatory measure.   We have added a sentence to the discussion about our intention to utilize fluorescent microscopy in future studies.

For the scientific point of view, it will be very interesting to know what was the immunological response of the eradication of the biofilms.

We agree that the immunological responses will be an important part of future evaluations of this material in infection prevention and treatment.  While we did not use an infected model in these preliminary studies, our results serve as a baseline for comparing such future work.   We have updated the discussion to address this future exploration.  

Reviewer 3 Report

The manuscript explores an important health topic biofilm related infections. The authors evaluate the potential of injectable mannitol-chitosan blended pastes loaded with antibiotics against S. aureus. The overall study design seems adequate but lacks a lot of detail hampering its replication by other groups and also raising a few questions.

Line 26. Keywords should be checked (biomaterial and infection are duplicated; Staphylococcus should be in italic)

Line 71. Results section: More detailed should be provided

Line 78. This is a general comment for all tables and figures. All Figures, and Tables should have a short explanatory title and caption. Using as example Figure 1 I would suggest the following: Figure 1. Enzymatic degradation results. Graph represents….

In graphs please check the axis title. For example if you are assessing a concentration over time use for Y axis the title Concentration (units) and for X axis Time (unit).

When statistically significant differences were found represent them in the graphs and refer to the p value. In table 1, line 79, the authors wrote “The ** indicates a statistical difference…..” but the p value is missing and is also not described in the material and methods section.

Line 103-104. How the authors explain the results presented for day 2 with such a huge error bar?

Lines 108-111. According to the authors criteria when the measured ZOI (that should be first mentioned as zone of inhibition or its meaning) average is bellow 0.2 mm was reported as 0. Nevertheless, the images presented in figure 4 are tricky. For example is difficult to observe a difference between ChMPEG (days 6, 7) and Ch-PEG day 5 specially because in this last case the blank disk is so close to the plate edge that even if a ZOI as large as those observed at day 1 that would not be visible. There is no space left to allow the appearance of a symmetric ZOI.

Line 121. This section should be better organized because it is not clear. In figure 5 the results of Ch-PEG no ABX (I would suggest replacing no ABX by without ABX or blank), Ch-PEG (loaded with a solution of 10mg/mL amikacin and vancomycin?), the same for ChMPEG and 2 concentrations of free ABX (mixture of antibiotics?). The authors conclude that all groups except Ch-PEG no antibiotics showed a decrease in bacterial viability. Was this decrease significant? Are the released concentrations of ABX from Ch-PEG and ChMPEC in the same range of the free antibiotics?

Here are the results of mannitol addition (lines 123-125) presented?

Line 131. Please add something because like this you are only presenting figure 6.

Lines154-229. Discussion could be improved

Line 230. Material and Methods. The all section should be improved in order to provide enough detail to reproduce the experiments. The controls used  should be described as well as suppliers. In addition I would made specific comments.

Line 232-235. Please reorganize the text because it is very difficult to read.

Line 254-263. Only at the end it is clear that different samples were used for each point. It would be better to start by saying this.

Later (line261) is written that samples were placed in an oven at 45ºC and after drying were weighed and compared to their initial weight. For how long were the samples allowed to dry? How was determined the initial weight? Should we assume that samples were dried to a constant mass as meant in a pharmacopeia method?

Line 264. Elution profiles and antibiotic activity should be describe in separate sections

Line 268. Explain what you meant by complete refreshment

Lines 268- 272. Describe the HPLC assay in such a way that it could be reproduced elsewhere, e.g. among other things is missing: mobile(s) phase(s), flow rate, detectors wavelengths, derivatization procedure, etc.

Line 274-280. The text could be re-written in order to increase clarity. In order to validate the assay negative and positive controls must have been performed and described here.

Lines 281-290. Evaluation of antimicrobial activity using MBEC is not conducted under static conditions. Please describe how you did it?

The concentrations of the antibiotics from the elution samples should be mentioned (line 285).

The conditions in which the sonication procedure is performed could affect bacteria viability. For this reason, stating that samples were sonicated for 5 minutes (line 287) is not enough. The apparatus used and the frequency should be described.

Describe the viability assay with Prest blue (288).

Line 291. I suggest that Cytocompatibility and biocompatibility should be splitted or at least mention in lines 292 and 301 that cytocompatibility and biocompatibility assays are described, respectively.

Line 292. The ATCC number or similar of the cell line should be introduced.

Line 292-299. Brieffly describe how cells were grown and how the Cell-Titer Glo viability assay was performed.

Lines 309- 311. Describe how histological blocks were prepared, sections were obtained and stained with hematoxylin eosin (H&E- that should be written in full).

Line 312. When where the results considered statistically significant?

Author Response

Review 3:

Comments and Suggestions for Authors

The manuscript explores an important health topic biofilm related infections. The authors evaluate the potential of injectable mannitol-chitosan blended pastes loaded with antibiotics against S. aureus. The overall study design seems adequate but lacks a lot of detail hampering its replication by other groups and also raising a few questions.

Line 26. Keywords should be checked (biomaterial and infection are duplicated; Staphylococcus should be in italic)

Changed in text.

Line 71. Results section: More detailed should be provided

Results section was updated to include specific values reported in the figures and tables.

Line 78. This is a general comment for all tables and figures. All Figures, and Tables should have a short explanatory title and caption.

Captions were added to all tables, and some titles were adjusted.

Using as example Figure 1 I would suggest the following: Figure 1. Enzymatic degradation results. Graph represents….

Captions were updated to follow this format.

In graphs please check the axis title. For example if you are assessing a concentration over time use for Y axis the title Concentration (units) and for X axis Time (unit).

Graph axes were updated to this format.

When statistically significant differences were found represent them in the graphs and refer to the p value.

In table 1, line 79, the authors wrote “The ** indicates a statistical difference…..” but the p value is missing and is also not described in the material and methods section.

This was a major oversight on our part. All statistical values were added throughout the text.

Line 103-104. How the authors explain the results presented for day 2 with such a huge error bar?

Discussion has been updated to reflect this large standard deviation.  There was one outlier with a higher amount than all others.  However, we have included this data point in the analysis and have provided possible reasons for this variability in the data. 

Lines 108-111. According to the authors criteria when the measured ZOI (that should be first mentioned as zone of inhibition or its meaning)average is bellow 0.2 mm was reported as 0. Nevertheless, the images presented in figure 4 are tricky. For example is difficult to observe a difference between ChMPEG (days 6, 7) and Ch-PEG day 5 specially because in this last case the blank disk is so close to the plate edge that even if a ZOI as large as those observed at day 1 that would not be visible. There is no space left to allow the appearance of a symmetric ZOI.

ZOI was spelled out. Images were updated. Statistical values were added to Table 2, which numerically describes the zones shown in the images in Figure 4. It may be difficult to see the zones for the later time points in the images, but we did measure zones through the ImageJ software.

Line 121. This section should be better organized because it is not clear. In figure 5 the results of Ch-PEG no ABX (I would suggest replacing no ABX by without ABX or blank), Ch-PEG (loaded with a solution of 10mg/mL amikacin and vancomycin?), the same for ChMPEG and 2 concentrations of free ABX (mixture of antibiotics?). The authors conclude that all groups except Ch-PEG no antibiotics showed a decrease in bacterial viability. Was this decrease significant? Are the released concentrations of ABX from Ch-PEG and ChMPEC in the same range of the free antibiotics?

The group titles were renamed for clarity. Statistics were added to indicate significant differences between groups. Values were selected based on the reported biofilm eradication concentrations.

Where are the results of the mannitol addition (lines 123-125) reported?

These lines were revised.

Line 131. Please add something because like this you are only presenting figure 6.

This figure was updated to include more information and statistics. Details about results for this section were also added.

Lines154-229. Discussion could be improved

Sections were added to give better depth to the discussion. If there are additional changes you would like to see in the discussion, please let us know.

Line 230. Material and Methods. The all section should be improved in order to provide enough detail to reproduce the experiments. The controls used should be described as well as suppliers. In addition I would made specific comments.

Details were added according to your specific comments. Suppliers and controls were added.

Line 232-235. Please reorganize the text because it is very difficult to read.

This sentence was rephrased for clarity.

Line 254-263. Only at the end it is clear that different samples were used for each point. It would be better to start by saying this.

Opening sentence now includes this statement.

Later (line261) is written that samples were placed in an oven at 45ºC and after drying were weighed and compared to their initial weight. For how long were the samples allowed to dry? How was determined the initial weight? Should we assume that samples were dried to a constant mass as meant in a pharmacopeia method?

Added the original mass for each sample and how many hours the samples were allowed to dry.

Line 264. Elution profiles and antibiotic activity should be describe in separate sections

These were split into two separate sections.

Line 268. Explain what you meant by complete refreshment

New PBS was added each day after samples were taken (changed in text).

Lines 268- 272. Describe the HPLC assay in such a way that it could be reproduced elsewhere, e.g. among other things is missing: mobile(s) phase(s), flow rate, detectors wavelengths, derivatization procedure, etc. 

All HPLC settings were added to this section.

Line 274-280. The text could be re-written in order to increase clarity. In order to validate the assay negative and positive controls must have been performed and described here. 

We included details about the positive control used. Negative controls were not used in this assay as unloaded discs would not prevent any zone of bacterial growth. Future studies can include negative controls.  

Lines 281-290. Evaluation of antimicrobial activity using MBEC is not conducted under static conditions.

Please describe how you did it?

This section was updated to include details about the orbital shaker used and rpm.

The concentrations of the antibiotics from the elution samples should be mentioned (line 285).

Concentrations chosen to be representative of concentrations of first day release in eluates.  This was added to this section with exact values.

The conditions in which the sonication procedure is performed could affect bacteria viability. For this reason, stating that samples were sonicated for 5 minutes (line 287) is not enough. The apparatus used and the frequency should be described.

Apparatus and frequency were added.  

Describe the viability assay with Prest blue (288).

A brief description of the prestoblue assay was added

Line 291. I suggest that Cytocompatibility and biocompatibility should be split or at least mention in lines 292 and 301 that cytocompatibility and biocompatibility assays are described, respectively.

Cytocompatibility and biocompatibility sections were split for clarity.

Line 292. The ATCC number or similar of the cell line should be introduced.

Cell line details were added.

Line 292-299. Briefly describe how cells were grown and how the Cell-Titer Glo assay was performed.

Cell growth procedure was described. Details about the Cell-titer glo assay were added.

Lines 309- 311. Describe how histological blocks were prepared, sections were obtained and stained with hematoxylin eosin (H&E- that should be written in full).

H&E was spelled out and preparation procedure was added.  

Line 312. When where the results considered statistically significant?

This information was added (P < 0.05).

Reviewer 4 Report

The manuscript presented by Pace et al. deals with the possibility to obtain injectable mannitol-chitosan paste for musculoskeletal infection prevention; this formulation holds intrinsic antimicrobial properties due to mannitol anti-biofilm properties. Moreover, the paste has been expolred as drugs carrier by using antibiotics (vancomycin and amikacin). 

The obtained mterials has been studied in terms of injectability, degradation and release profile, while biological properties have been tested against the pathogen S. aureus and cytocompatibility has been verified in vitro and in vivo.

In general, the paper is clear and well presented; the topic is of interest for the Journal's readers and the results are supported by a suitable experimental design.

Accordingly, I consider this manuscript as worthy of consideration for publication after some minor revisions; here my comments.

-Figures: in the file available for revision figures are very blurry and pale, please ensure to provise better quality images.

-Figure 3: the standard deviation seems to be very high (particulary for day 2). Is it due to a poor homogeneity  of the samples or due to technical limitations?

-Figure 4: it looks to me that the inhibition halo can be appreciate only until day 4, afterwards I have some doubts that the results can be considered as significant.

Figure 6: any significant difference between the groups?

Author Response 

Review 4:

Comments and Suggestions for Authors

The manuscript presented by Pace et al. deals with the possibility to obtain injectable mannitol-chitosan paste for musculoskeletal infection prevention; this formulation holds intrinsic antimicrobial properties due to mannitol anti-biofilm properties. Moreover, the paste has been explored as drugs carrier by using antibiotics (vancomycin and amikacin). 

The obtained materials has been studied in terms of injectability, degradation and release profile, while biological properties have been tested against the pathogen S. aureus and cytocompatibility has been verified in vitro and in vivo.

In general, the paper is clear and well presented; the topic is of interest for the Journal's readers and the results are supported by a suitable experimental design.

Accordingly, I consider this manuscript as worthy of consideration for publication after some minor revisions; here my comments.

Thank you for your time and for your helpful comments.

-Figures: in the file available for revision figures are very blurry and pale, please ensure to provide better quality images.

Images were updated and added at higher resolution.

-Figure 3: the standard deviation seems to be very high (particulary for day 2). Is it due to a poor homogeneity of the samples or due to technical limitations?

Discussions section was updated to address this concern.  There was one outlier with high concentration, but we have included in the analysis and have added potential reasons and ways to address this in future studies

-Figure 4: it looks to me that the inhibition halo can be appreciate only until day 4, afterwards I have some doubts that the results can be considered as significant.

The images used were only representative of results, which were reported in the table.  The ZOI study was primarily a confirmatory measure for the activity of antibiotics eluted. Some images used in this figure have been updated for clarity.

Figure 6: any significant difference between the groups?

This figure was updated to indicate where a significant difference was seen.

Round 2

Reviewer 2 Report

I can see the efforts of authors for the MS. However, it seems that the authors are in a hurry to get the paper published. I still have some minor concerns about the MS and want authors attention please see a few suggestions. 

Review 2:

Comments and Suggestions for Authors

In the MS entitled “Injectable mannitol chitosan blended past eradicates S. aureus biofilms and elutes antimicrobials.” Pace et al has shown the role of Ch-PEG and ChMPEG as a biodegradable adjuvant therapy for musculoskeletal infection prevention for Surgical Site Infection (SSI). I have many concerns with this MS. The MS is at quite an immature stage, the MS is poorly written and lacks basic information as if how the experiments are done.

Thank you for your time and for your helpful comments. This is an ongoing study reporting material characteristics determined by in vitro and in vivo evaluations for this new paste material. Future and ongoing expansion includes additional in vivo testing, including in infected models. Would you suggest that we add “exploratory” into the introduction or the title to clarify that this text reports preliminary results? The description of all methods have been updated to include more detail about how the experiments were conducted.

Concern: The title is still misleading and confusing..:(

How about:

Characterization and anti-biofilm activity of mannitol-chitosan blended paste for local antibiotic delivery system.

Abstract:

What is moderate inflammatory response? Authors should mention how they came to this conclusion (p-value?).

The text was updated to include more detail about the moderate inflammatory response conclusion based on a 5 point histological scale. The p value was added to this sentence of the abstract. We cannot reference the study we used to determine inflammatory response rankings within the abstract, but it has been referenced in the methods.

Authors should have checked the standard recommended by Graphpad prism software for p value

Symbol   Meaning

Ns P > 0.05

* P ≤ 0.05

**P ≤ 0.01

***P ≤ 0.001

****P ≤ 0.0001 (For the last two choices only)

The one star mentioned by the authors does not justify the obtained p value

At the end of the abstract authors claims that mannitol paste with or without antibiotics decreases bacterial viability, I am surprised by this conclusion, do authors claim that there is no role of antibiotics over S. aureus and mannitol paste can take place of antibiotics. The statement should be rewritten or explained properly.

To be clear, we did not develop this mannitol paste to replace antibiotics; however, the observed activity is promising for infections that are not covered by some of the common antibiotics. The intent of this paste was specifically to be used in conjunction with aminoglycosides to take advantage of the effect reported by Allison et al. This local delivery system is intended as an adjunct therapy for patients with trauma or surgical procedures who would likely be on systemic antibiotic.

The description in the MS does not specify the findings or intent of the authors. The data and description is misleading

Introduction:

Lack of literature cited, there is lots of literature about the role of metabolites and biofilm dispersion. Authors need to cite the latest known facts about this important aspect of biofilm research.

Citations in the introduction have been updated to reflect the most recent advances in biofilm research.

Results:

Results are encouraging but the not concluded in a scientific way, I strongly believe that the paper should be retitled or more emphasis should be given on S. aureus biofilms or mixed biofilms. Authors are trying to prove that Ch-PEG is better than ChMPEG but not through the activity against biofilm but by their enzymatic degradation.

We chose to include S. aureus in the title because we only report results from that strain. To clarify about degradation, we are reporting the enzymatic degradation of the material, not enzymatic degradation of biofilm. Furthermore, we are not concluding that the Ch-PEG paste is “better”, but simply reporting that one material degrades in the body faster. Could you provide clarification on how the title should change? One alternative we considered is “Injectable mannitol-chitosan blended paste local antibiotic delivery system: material characterization and anti-biofilm activity”.

I do not understand the concept of using just syringe and syringe with 18 G needle, what is the takeaway message, how authors plan to inject paste in a patient just with the syringe. It’s confusing and needs to be addressed.

Eventually, we would like a delivery system that is applicable in complex wounds or non-surgical delivery in minimally invasive manner, which is why we investigated its injectable potential through a syringe. We removed the results for the 18G needle, as it does not support this application.

Figure 1 – Figure indicates Ch-PEG is more degradable then CHMPEG, where is the statisticaldata? No p-value? How the conclusion was reached?

P value and asterisks were added.

Figure 2 – Same problem in antibiotic and mannitol elution. It’s not clear from the Materials and Methods sections how the conclusion was made.

Statistical values were added to these graphs as well.

Figure 4 – I personally cannot see any difference in the antimicrobial activity against S. aureus, why none of the figures have statistical data?

Statistical values were added to Table 2, which numerically describes the zones shown in the images in Figure 4. It may be difficult to see the zones for the later time points in the images, but we did measure zones through the ImageJ software.

Figure 5 – Data indicates that there is not any role of antibiotics in biofilm eradication. Though antibiotic, it shows almost 40% of bacterial eradication. How this was tested? Plate assay? Viability assay?

We are not claiming that antibiotics have no role—they are certainly capable of eradicating biofilm at very high concentrations. A delivery system is capable of achieving these high concentrations locally and would be used as an adjunctive therapy. This test was to evaluate whether the paste materials were able to elute enough mannitol in combination with antibiotics to eradicate biofilm. It was unexpected that non-antibiotic loaded groups reduced viability to the degree that they did. As stated in the methods section, this was a viability assay using Prestoblue to determine bacteria viability.

Figure 7 – Biocompatibility assay – The magnification of the H&E image is too low to see if there is any kind of immune response, in forms of granulocytes. Authors must zoom in to see if there is a migration of neutrophils.

We have added images at 20x magnification for representative areas at the periphery of the material, which shows granulocytes near and migrating into the implant.

How thick is the section?

The section thickness is 5 μm. The text has been updated to include this value.

Figure 8 should be Figure 7 (b) and no statistical data shown for the histological scores.

Figure 8 was changed to figure 7b. Statistical data was added to indicate no significant difference between groups (p>0.05).

It will be helpful for the readers of the MS, if more details about how the MS is grown and how the eradication studies were done, some fluorescent microscopy pictures will help to understand the impact of eradication.

We agree that fluorescent images are valuable for illustrating bacterial biofilm characteristics. However, we did not have access to a microscope with high enough magnification and resolution to use this as a confirmatory measure. We have added a sentence to the discussion about our intention to utilize fluorescent microscopy in future studies.

For the scientific point of view, it will be very interesting to know what was the immunological response of the eradication of the biofilms.

We agree that the immunological responses will be an important part of future evaluations of this material in infection prevention and treatment. While we did not use an infected model in these preliminary studies, our results serve as a baseline for comparing such future work. We have updated the discussion to address this future exploration.

Author Response

Comments and Suggestions for Authors

Concern: The title is still misleading and confusing..:(

How about:

Characterization and anti-biofilm activity of mannitol-chitosan blended paste for local antibiotic delivery system.

Thank you so much for your suggestion. We have changed the title to “Characterization and anti-biofilm activity of mannitol-chitosan blended paste for local antibiotic delivery system.”

Abstract:

What is moderate inflammatory response? Authors should mention how they came to this conclusion (p-value?).

The text was updated to include more detail about the moderate inflammatory response conclusion based on a 5 point histological scale. The p value was added to this sentence of the abstract. We cannot reference the study we used to determine inflammatory response rankings within the abstract, but it has been referenced in the methods.

Authors should have checked the standard recommended by Graphpad prism software for p value

Symbol   Meaning

Ns P > 0.05

* P ≤ 0.05

**P ≤ 0.01

***P ≤ 0.001

****P ≤ 0.0001 (For the last two choices only)

The one star mentioned by the authors does not justify the obtained p value

We have incorporated a similar type of symbol system for most of the figures in the text, while also adding the obtained p values in the figure and table descriptions. We have also included the type of statistical test used in each of the table and figure descriptions as well.

At the end of the abstract authors claims that mannitol paste with or without antibiotics decreases bacterial viability, I am surprised by this conclusion, do authors claim that there is no role of antibiotics over S. aureus and mannitol paste can take place of antibiotics. The statement should be rewritten or explained properly.

To be clear, we did not develop this mannitol paste to replace antibiotics; however, the observed activity is promising for infections that are not covered by some of the common antibiotics. The intent of this paste was specifically to be used in conjunction with aminoglycosides to take advantage of the effect reported by Allison et al. This local delivery system is intended as an adjunct therapy for patients with trauma or surgical procedures who would likely be on systemic antibiotic.

The description in the MS does not specify the findings or intent of the authors. The data and description is misleading

We have rewritten the abstract to include more detail about the intent and specific results (degree of response).  This has been patterned to follow the order of presentation within the manuscript.  Edits throughout the manuscript have also been incorporated to clarify descriptions of data to present numerical results in addition to figures.

Reviewer 3 Report

The quality of the manuscript did not improved by the author's replies. The explanations provided reveal several major flaws, such as missing (antibacterial activity) or inadequate (using tissue culture plastic as control is incorrect, untreated cells are the correct control) controls for several experiments, references are missing and lack of optimization of manufacturing process. When one suspects of lack of homogeneity in a product instead of including the so-called outlier the process should be optimized and then performed independent experiments with at least triplicates to get consistent results. The authors were questioned about the huge SD for day 2 they reply with the outlier answer but they did not explain for example the huge variability for day 4.

The results of the statistical analysis are still not clearly presented through the all document.

The figures and tables are still not well organized. For example in table one the title of the second column is not correct.  If we are comparing forces, force instead of 1mL syringe should be the title.

Author Response

The quality of the manuscript did not improved by the author's replies. The explanations provided reveal several major flaws, such as missing (antibacterial activity) or inadequate (using tissue culture plastic as control is incorrect, untreated cells are the correct control) controls for several experiments, references are missing and lack of optimization of manufacturing process. When one suspects of lack of homogeneity in a product instead of including the so-called outlier the process should be optimized and then performed independent experiments with at least triplicates to get consistent results. The authors were questioned about the huge SD for day 2 they reply with the outlier answer but they did not explain for example the huge variability for day 4.

Thank you for your persistence in assisting with the revision, and we realize that some of the descriptors did not convey the actual setup. 

 For all of the antibacterial activity assays, including the ZOI study and the paste eluate study with MBEC plates, blank controls and controls including several standard antibiotic concentrations were used (as an internal check of the assay). However, these were not reported for brevity (blanks were always 0, and standards were the same for each time point).  However, the materials and methods have been updated to better reflect what was done and any confusing wording has been corrected. One instance in particular is the use of “tissue culture plastic” which was colloquially meant to indicate wells with untreated, “blank paste,” cells.

We have revisited the HPLC method that was used and identified and optimized a method to re-test eluates from this study as well as several technical replicates.  To address the comment regarding the “lack of optimization of manufacturing process,” results for the mannitol elution profile have been rerun on several replicates, with consistent results for multiple batches, including batches where the mannitol paste was loaded with and without antibiotics. These two independent elution studies (with and without antibiotics) were done at a different time points than the study reported in the text and were done using a different batch of mannitol paste material. The HPLC method, could have been to blame for the outlier first reported.  We have updated the materials and methods, the results, and other sections accordingly.

The results of the statistical analysis are still not clearly presented through the all document.

We have modified the symbols used to indicate statistical difference and have included all of our statistical methods in the material and methods section. We have also included the different statistical methods used for each table and figure in their respective titles.

The figures and tables are still not well organized. For example in table one the title of the second column is not correct.  If we are comparing forces, force instead of 1mL syringe should be the title.

This table has been changed to include the correct title. Other figures and tables have been checked and corrected as appropriate. 

Round 3

Reviewer 3 Report

The authors made an effort in improving the manuscript but:

According to the journal instructions "Abstract: The abstract should be a total of about 200 words maximum" and in its present version the abstract has more than 300 words (in my count 371). Table 1. Assuming that N is the force unit, consider Force (N) and remove all the N Figure 3 data improved and changed a lot since from day 4 no compound was detected. It is not clear if the data presented is an average of different experiments or one representative set. This is important because the authors claim that they use different time points but this is not described in the methods section. Including the controls and its results is not a repetition. In contrary is important to evaluate the adavantage of the pastes developed by the authors over the currentely available antibiotics. How can the authors garantee that only manitol realease has changed? The antibiotic realease was not evaluated neither all the other data.

Author Response

We thank the reviewer again for the time and thoroughness toward improving the clarity of the manuscript content.

According to the journal instructions "Abstract: The abstract should be a total of about 200 words maximum" and in its present version the abstract has more than 300 words (in my count 371).

Additional data and explanations had been added at the request of another reviewer, which overshot the word limit.  We have now edited the abstract to 197 words, while trying to keep information that conveys intent of the study and summarizes the major findings. 

Table 1. Assuming that N is the force unit, consider Force (N) and remove all the N

Thank you for pointing this out--this has been revised in the manuscript.

Figure 3 data improved and changed a lot since from day 4 no compound was detected. It is not clear if the data presented is an average of different experiments or one representative set. This is important because the authors claim that they use different time points but this is not described in the methods section. Including the controls and its results is not a repetition. In contrary is important to evaluate the adavantage of the pastes developed by the authors over the currentely available antibiotics. How can the authors garantee that only manitol realease has changed? The antibiotic realease was not evaluated neither all the other data. 

This is one representative set.  We retested both the original samples as well as another identical elution study that was conducted to evaluate biofilm eradication (more sample was required than we had left from the original study).  This procedure which followed the identical methods described in the methods section is not a different “time point” but a technical replicate. The results for mannitol elution compared between the two, using the updated HPLC protocol, are very similar.  We now have very little of the original sample left. 

We did confirm that vancomycin elution followed similar patterns to the set of samples originally tested, releasing detectable quantities for 7 days. Retesting the amikacin released requires additional time for precolumn derivatization and we cannot perform this within the timeframe provided by the editor.  However, we are confident that the elution between these technical replicates is similar. 

As far as evaluating the advantages of this paste over the non-mannitol blends, we feel that our results for eradication of biofilm are pertinent and that extending elution is advantageous in infection treatment.  We reiterate that we are not suggesting this drug delivery system to replace antibiotics, but rather as an adjunct and delivery system for them.  Advantages of this system compared to other local drug delivery systems, including other injectable delivery systems, is provided in the intro and discussion. This manuscript reports the characterization and efficacy studies, so that expanded studies can further evaluate the advantages for infection prevention.